# Metabolite and Transcriptome Profiles of Proanthocyanidin Biosynthesis in the Development of Litchi Fruit

**DOI:** 10.3390/ijms24010532

**Published:** 2022-12-28

**Authors:** Ruihao Zhong, Junbin Wei, Bin Liu, Honghui Luo, Zhaoqi Zhang, Xuequn Pang, Fang Fang

**Affiliations:** 1State Key Laboratory for Conservation and Utilization of Subtropical Agro-Bioresources/Guangdong Provincial Key Laboratory of Postharvest Science of Fruit and Vegetables/Engineering Research Center for Postharvest Technology of Horticultural Crops in South China, South China Agricultural University, Guangzhou 510642, China; 2College of Life Sciences, South China Agricultural University, Guangzhou 510642, China; 3College of Horticulture, South China Agricultural University, Guangzhou 510642, China

**Keywords:** litchi, proanthocyanidin biosynthesis, transcriptomic analysis, gene screening, gene expression regulation

## Abstract

The fruit of *Litchi chinensis* contains high levels of proanthocyanidins (PAs) in the pericarp. These substances can serve as substrates of laccase-mediated rapid pericarp browning after the fruit is harvested. In this study, we found that the major PAs in litchi pericarp were (−)-epicatechin (EC) and several procyanidins (PCs), primarily PC A2, B2, and B1, and the EC and the PC content decreased with the development of the fruit. RNA-seq analysis showed that 43 early and late structure genes related to flavonoid/PA biosynthesis were expressed in the pericarp, including five *ANTHOCYANIDIN REDUCTASE* (*ANR*), two *LEUCOANTHOCYANIDIN REDUCTASE* (*LAR*), and two *ANTHOCYANIDIN SYNTHASE* (*ANS*) genes functioning in the PA biosynthesis branch of the flavonoid pathway. Among these nine PA biosynthesis-related genes, *ANR1a*, *LAR1*/*2,* and *ANS1* were highly positively correlated with changes in the EC/PC content, suggesting that they are the key PA biosynthesis-related genes. Several transcription factor (TF) genes, including MYB, bHLH, WRKY, and AP2 family members, were found to be highly correlated with *ANR1a*, *LAR1*/*2,* and *ANS1*, and their relevant binding elements were detected in the promoters of these target genes, strongly suggesting that these TF genes may play regulatory roles in PA biosynthesis. In summary, this study identified the candidate key structure and regulatory genes in PA biosynthesis in litchi pericarp, which will assist in understanding the accumulation of high levels of browning-related PA substances in the pericarp.

## 1. Introduction

Litchi (*Litchi chinensis* Sonn.) is a characteristic tropical to subtropical fruit crop. Litchi fruit is popular with consumers because of its bright red skin, sweet juicy flesh, and high nutritional value. However, the litchi pericarp readily undergoes browning once the fruits have been detached from the tree, which significantly shortens their storage life and reduces their commercial value [1]. Pericarp browning has long been regarded as the main postharvest problem relating to the fruit [2]. As a consequence, the mechanism of pericarp browning has always been a key topic of interest in research concerning the postharvest issues of litchi fruit, in which the analysis of the key browning-associated enzymes and substrates represents a critical and challenging problem.

Litchi pericarp browning has long been ascribed to enzymatic browning by polyphenol oxidases (PPOs) [1]. However, we found that an anthocyanin degradation-related laccase (anthocyanin degradation enzyme/laccase, ADE/LAC) was abundant in the pericarp and was very active in the co-oxidization of (−)-epicatechin (EC) and anthocyanins, leading to the degradation of anthocyanins and pericarp browning [3,4]. Recently, we demonstrated that ADE/LAC purified from litchi pericarp was able to catalyze the oxidative polymerization of EC and procyanidin (PC) A2/B2/B1 and produced brown polymerization products in vitro [5]. EC, (+)-catechin (CT) and their polymers (PCs) belong to proanthocyanidins (PAs), also called condensed tannins, which are specific secondary metabolites in the plant kingdom. Pas play multiple roles in plants and, in particularly, provide important protection from biotic and abiotic damage [6]. Recently, the values to human health of Pas, such as their antioxidant, anticancer, and cardiovascular disease alleviation properties, have attracted wide interest among researchers [7]. The oxidative polymerization of Pas leading to tissue browning has also been observed in various other plant species. For example, the oxidative polymerization of EC/CT and PCs was reported to be related to seed coat browning in *Brassica napus* [8]. In red rice (*Oryza sativa*), the oxidation of PCs is one of the factors that causes color deepening [9]. An *Arabidopsis thaliana* mutant (*tt10*) that is deficient in AtLAC15 fails to develop a normal brown seedcoat during seed maturation, retaining levels of EC and oligo PCs in the seed coat after maturation that are significantly higher than the wild-type, suggesting that LAC-mediated EC/PC oxidative polymerization is associated with seed coat browning [10]. Accordingly, we consider that the LAC-mediated oxidative polymerization of PAs plays an important role in litchi pericarp browning. Among the PA substances, EC was identified as the most favored natural substrate in the litchi pericarp [5].

PA substances are the major phenolics of the litchi pericarp, representing more than 80% of the phenolic compounds [11,12,13]. Numerous studies have shown that the pericarp contains high levels of EC and its oligomers (e.g., PC A2 and B2), with a total PC content that is around 10 times higher than that of an apple peel [14,15]. It was found that the EC and PC content in litchi pericarp declined with postharvest storage or browning [12,16], suggesting that they may be involved in the postharvest browning of the pericarp [13]. Our previous studies also found that the EC and PC content decreased as the pericarp browned, and ADE/LAC showed high activity in relation to these substances [4,5,17]. Therefore, we suggest that EC and PCs are the major browning substrates during litchi pericarp browning and play an important role in the browning process. Exploring the accumulation mechanism of the PA substances (mainly EC and PCs) will be valuable for understanding the rapid pericarp browning of litchi fruit after harvest. Although the biosynthesis pathway of PAs in plants has been intensively studied, there are few reports on PA biosynthesis and its regulation in litchi fruit, which has a particularly high PA content.

The biosynthesis of PAs shares the phenylalanine-chalcone-leucoanthocyanidin pathway with anthocyanin biosynthesis [18], which has been identified as a step-based process involving phenylalanine ammonia-lyase (PAL), cinnamate-4-hydroxylase (C4H), 4-coumarate: CoA ligase (4CL), chalcone synthase (CHS), chalcone isomerase (CHI), flavanone 3-hydroxylase (F3H), flavonoid3′-hydroxylase (F3′H), and dihydroflavonol reductase (DFR). At the level of the (leuco)anthocyanidin, PA biosynthesis branches from the anthocyanin pathway, leading to the production of epi (catechin) under the function of anthocyanidin synthase (ANS), anthocyanidin reductase (ANR), and leucoanthocyanidin reductase (LAR) (Figure 1). LAR catalyzes the conversion of leucocyanidin to the CT [19], whereas ANS converts leucocyanidins to cyanidins, which are consequently reduced to EC by ANR [20]. Therefore, LAR and ANR are usually considered to be the specific enzymes for the PA branch of the flavonoid biosynthesis pathway [18,21,22,23,24]. Recently, the whole genome sequence of litchi was published [25], in which multiple members of the *LAR*, *ANR,* and *ANS* gene families were annotated. However, the key genes involved in PA biosynthesis in litchi pericarp have not yet been identified.

Recently, studies on the molecular regulation of the flavonoid/PA biosynthesis pathway have focused on mining the regulatory genes of the pathway [26,27]. Many transcription factors (TFs), such as the MYB, bHLH, and WD40 gene family members, have been shown to regulate the PA pathway-related genes, *ANR* and *LAR* [28,29]. The MYBs that specifically regulate *ANR* and *LAR* have been functionally characterized for Arabidopsis (AtTT2) [30], grapevine (VvMYBPA1 and VvMYBPA2) [31,32], persimmon (DkMYB2 and DkMYB4) [33,34], poplar (PtMYB134) [35], and *Medicago truncatula* (MtPAR) [36]. However, few studies have reported on the identification and characterization of the genes regulating the PA biosynthesis pathway in litchi.

The integrated analysis of metabolite profiles and transcriptomes has recently been used as an important tool for screening the genes that function in the metabolite biosynthesis pathway. In this study, we first analyzed the changes in the PA content of litchi pericarp during fruit development. Then, we profiled the transcriptome changes in the litchi pericarp at different development stages to investigate the structural and regulatory genes associated with PA biosynthesis. The study provides valuable insight into the accumulation of the browning substrates that will be of assistance in further research regarding the mechanisms of litchi pericarp browning.

## 2. Results

### 2.1. Changes in the Proanthocyanidin and Anthocyanin Content of the Pericarp during Fruit Development

To understand the accumulation of proanthocyanidins (PAs) in the pericarp during litchi fruit development, pericarp tissues were sampled at 30, 60, 75, and 85 days after female flower anthesis (DAF) (Figure 1A) and used for HPLC analysis. PA substances, primarily (−)-epicatechin (EC) and procyanidins (PCs) A2, B2, B1, as well as anthocyanins, were well separated by the HPLC chromatographic conditions, and identified by a comparison with the retention times and absorption spectra of the respective standards (Figure 1B,C, Appendix A). The identification of these PAs was further confirmed by the similar chromatography profiles of the PA substances detected for litchi pericarp [37], although co-elution with other minor PCs could not be unambiguously ruled out. The contents of these substances were estimated according to their standard curves (Appendix A). EC was the most abundant PA in litchi pericarp, followed by PC A2 and B2. Large amounts of EC (66.9 ± 3.9 mg g^−1^ DW) and PCs (109.5 ± 8.4 mg g^−1^ DW as the total PC content, including EC and the PCs) were detected in the pericarp during the young fruit period (30 DAF). The EC, total PC, and PC content decreased as the fruit developed (Figure 1D), although, after fruit maturation at 85 DAF, it still maintained around one-third of the content observed at the young fruit stage (30 DAF). Because almost no (+)-catechin was detected in the pericarp, we focused on (−)-epicatechin (EC) and total PC content in our analysis described below. The pericarp of the litchi fruit turned red at 75 DAF and were fully red at 85 DAF (the fully mature stage) (Figure 1A). No anthocyanins were detected in the young fruit pericarp (30 DAF and 60 DAF), whereas a low content of cyanindin-3-rutinside (Cy3R) was detected at 75 DAF, and the highest content was observed at 85 DAF, correlating to the color-turning pattern of the fruit.

### 2.2. Analysis of Transcriptome Sequencing Data of Litchi Pericarp during Fruit Development

To explore the differences in gene expression correlating with changes in PA content during litchi fruit development, total RNA was extracted from the litchi pericarp samples at 30, 60, 75, and 85 DAF for RNA sequencing. The cDNA libraries constructed from the RNA samples were sequenced using the Illumina Hiseq platform. After removing low-quality short sequences, each library produced 35.6 to 43.4 million clean reads, and these were used for assembly. The clean data for each sample were approximately 5.4 Gb to 6.6 Gb. The Q30 percentages (sequencing error rate < 1%) obtained from 12 libraries were 93.79% to 94.70%, and the guanine-cytosine (GC) percentages of sequenced data from the 12 libraries ranged from 46.38% to 49.28%. All clean reads were compared to the reference genome of *Litchi chinensis* (http://121.37.229.61:82), with the mapping rate ranging from 88.82% to 90.26%. The results showed that the accuracy and quality of the sequencing data were satisfactory for further analysis. A summary of these sequencing results is presented in Appendix A.

A principal component analysis (PCA) was employed to identify expression profile differences between samples. The PCA score plot results clearly showed that data from the four developmental stages were well separated and formed four clusters (Figure 2A), with the first principal component (PC1), as 60.5% of the total variables, indicating that the expression patterns of the genes in the samples at 30, 60, 75, and 85 DAF were significantly different. In addition, the correlation of sample replicates was very high, which indicated that the transcriptome data were repeatable and reliable.

Differentially expressed genes (DEGs) across the four development stages (30 DAF vs. 60 DAF, 60 DAF vs. 75 DAF, 75 DAF vs. 85 DAF) were assessed via pairwise comparisons of the expression levels of the unigenes based on their RPKM values obtained using RNA-seq. A total of 7319 genes were found to be significantly differentially expressed in the comparisons between any two of the four development stages (Figure 2B). A total of 3631 DEGs were detected between 30 DAF and 60 DAF, with 1953 downregulated and 1678 upregulated; 3910 DEGs occurred between the 60 DAF and 75 DAF libraries, including 2435 downregulated and 1475 upregulated; and 2846 DEGs were detected between 75 DAF vs. 85 DAF, with 1850 downregulated and 996 upregulated. These results suggest that the expression of a large number of genes in litchi pericarp is influenced by the development stage, and in particular for the development from 60 to 75 DAF, which had the most abundant DEGs (Figure 2B,C). The Venn diagram shows the unique and shared DEGs of the different groups. We observed an overlap of 1511, 1172, and 797 DEGs between the 30DAF vs. 60 DAF and 60 DAF vs. 75DAF group, the 60 DAF vs. 75DAF and the 75 DAF vs. 85 DAF group, and the 30DAF vs. 60 DAF and 75 DAF vs. 85 DAF group, respectively (Figure 2B), with 412 DEGs shared by all of the groups. These results suggest that the expression of a large number of the genes changed as the fruit development stages progressed.

### 2.3. GO and KEGG Enrichment Analysis of the DEGs

To obtain an overall understanding of all the unigene sequence function information, the assembled transcriptome data were compared and annotated in the GO and KEGG pathway databases. The DEGs were assigned to three main categories in the GO enrichment analysis: a comparison of 30 DAF vs. 60 DAF, 60 DAF vs. 75 DAF, and 75 DAF vs. 85 DAF. The GO functional annotation indicated that DEGs were divided into 45 functional groups with biological processes (21 terms), cellular components (13 terms), and molecular functions (11 terms). The DEGs generated by the three groups showed a certain similarity in their GO functional classification patterns (Appendix A). In the biological process category, most DEGs were classified as metabolic processes and cellular processes, followed by single-organism processes. In the cell component category, the cell and cell part were relatively dominant, followed by the organelles and membrane groups. Of the molecular functions, catalytic activity and binding were the most representative.

To further investigate the influence of the DEGs on metabolic pathways, KOBAS 2.0 software was used for the KEGG (Kyoto Encyclopedia of Genes and Genomes) enrichment analysis to clarify the enrichment in biological function of each comparison group (30 DAF vs. 60 DAF, 60 DAF vs. 75 DAF, and 75 DAF vs. 85 DAF). The bubble diagram illustrates the top 20 enriched pathways with low corrected q-values. In the 30 DAF vs. 60 DAF comparison, the DEGs related to flavonoid biosynthesis (ko00941) and phenylpropane biosynthesis (ko00940) were predominantly enriched (Figure 3A); in the 60 DAF vs. 75 DAF comparison, the DEGs functioning in phenylpropane biosynthesis (ko00940) and the biosynthesis of secondary metabolites (ko01110) were predominantly enriched (Figure 3B); and in the 75 DAF vs. 85 DAF comparison, photosynthesis (ko00195) and ribosome-related DEGs (ko03010) were predominantly enriched (Figure 3C). Therefore, the KEGG enrichment analysis of DEGs showed that the phenylpropanoid and flavonoid biosynthesis pathways were the most significant and rich factors during the young litchi fruit development stages, and this may be correlated with the active biosynthesis of EC and PCs.

### 2.4. Expression of Flavonoid/PA Biosynthesis-Related Genes in Litchi Pericarp during Fruit Development

According to the GO and KEGG analyses, 43 genes encoding the enzymes in the flavonoid/PA biosynthesis pathways were found to be expressed in the pericarp during fruit development. These 43 genes encode the enzymes PAL, 4CL, CHS, CHI, F3H, F3′H, FLS, DFR, ANS, LAR, ANS, and UFTG. (Figure 4B).

To identify the potential key genes functioning in the accumulation of EC/PCs in litchi pericarp during fruit development, we conducted correlation analyses between the gene expression patterns and the change in the EC/PC content. Of the 43 flavonoid/PA biosynthesis pathway-related genes, 29 were significantly correlated with both the EC and total PC content, with 25 positively and four negatively correlated genes. Two *ANS*, two *LAR,* and five *ANR* genes were identified based on the KEGG annotation. We nominated these genes based on phylogenetic analysis (see below). Of these PA biosynthesis pathway-specific genes, *ANR1a* (LITCHI029356), *LAR1* (LITCHI023217), *LAR2* (LITCHI005474), and *ANS1* (LITCHI022925) were highly expressed in the early young fruit stage and downregulated with fruit development, which significantly correlated to the change in the EC/PC content (Figure 4C). One *UFGT* (LITCHI002457) and one *ANR* (LITCHI026910) were upregulated after 75 DAF, significantly correlating with the accumulation of anthocyanins. Overall, we identified nine PA biosynthesis pathway-specific genes, of which *ANR1a*, *LAR1*/*2,* and *ANS1* were highly correlated with the accumulation of EC/PCs.

### 2.5. Verification of the Expression of the Genes Involved in Catechin Biosynthesis

To verify the transcription data acquired using RNA-seq, the expression levels of nine PA biosynthesis-related genes (*ANR*, *LAR*, *ANS*) were detected using qRT-PCR. As shown in Figure 5A, the expression patterns of the genes determined using qRT-PCR data were highly consistent with the transcriptome analysis, supporting the reliability of the RNA-seq data. In addition, of the nine genes, the expression of four (*ANR1a*, *ANS1,* and *LAR1*/*2*) showed similarly decreasing patterns with fruit development, which was closely correlated with the PA content (Figure 5B). These results suggest that these genes may be the key genes with more important roles in PA biosynthesis than other genes.

### 2.6. Prediction of the Biochemical Characteristics of ANR, LAR and ANS Proteins

Sequence feature analysis showed that the theoretical isoelectric points (IP) of deduced ANR, LAR, and ANS proteins ranged from 5.35 to 8.87. The molecular weights of the nine selected proteins ranged from 30.18 kDa to 40.56 kDa and the length varied between 281 and 357 amino acids (Table 1). It is likely that all of the ANR, LAR, and ANS proteins were cytosol proteins, none of which were predicted to contain potential transit signal peptides. Additionally, the presence of variable N-glycosylation sites was predicted in all nine proteins, indicating the possible presence of post-translational modifications in these proteins.

### 2.7. Phylogenetic Trees of Litchi ANR, LAR and ANS Proteins

To examine the phylogenetic relationships of the litchi ANR, LAR, and ANS proteins with some relevant characterized proteins from other plant species, neighbor-joining (NJ) phylogenetic trees were constructed using MEGA5.0 based on full-length deduced protein sequences of *ANR*, *LAR,* and *ANS* genes (Figure 6). We nominated these genes based on the phylogenetic analysis. LcANR1a and LcANR1b were very similar, as were LcANR2a and LcANR2b. Compared to LcANR2a/b, LcANR1a/b and LcANR3 were closer to some characterized ANRs, such as AtANR/ BANYULS (*Arabidopsis thaliana*), MtANR (*Medicago truncatula*), and VvANR (*Vitis vinifera*). LcLAR1 more closely resembled GrLAR1 (*Gossypium raimondii*) and CsLARa/b (*Camellia sinensis*) than the litchi LAR, LcLAR2. LcANS1 was very similar to TcANS (*Theobroma cacao*) and was clustered in a major group with the ANSs from other species, such as PtANS1/2 (*Populus trichocarpa*) and VvANS (*Vitis vinifera*).

### 2.8. Screening of Potential Catechin Biosynthesis-Related Transcription Factor (TF) Genes

To understand the regulation of PA biosynthesis genes, we selected three PA biosynthesis pathway-related genes that showed highly correlated EC/PC content, namely *ANR1a*, *LAR1,* and *ANS1*, as target genes for the screening of their potential regulation TF genes. We carried out a coexpression analysis of the target genes and the TF genes that expressed during fruit development (Figure 7A). The top 20 TF genes that correlated (r^2^ > 0.993) to the *ANR1a*, *LAR1,* and *ANS1* genes were members of the MYB (v-myb avian myeloblastosis viral oncogene homolog), bHLH (Basic helixloop-helix), WRKY (N-terminal conservative sequence WRKYGQK), C2H2 (zinc-finger transcription factor), SBP (squamosa promoter binding protein), and NAC (NAM, ATAF, and CUC2) gene families. We also conducted a PlantCARE analysis and identified six types of cis-regulatory element related to the abovementioned TFs in the promoter regions (2000 bp upstream of the start codons) of the *ANR1a*/*LAR1*/*ANS1* genes (Figure 7B). We found that the promoters of *ANR1a*, *LAR1,* and *ANS1* contained three, three, and four MYBR (MYB-recognizing) elements, respectively, and had four, five, and one MYC (bHLH-binding) elements, respectively. These results suggest that MYB and bHLH may be the key regulators for the catechin biosynthesis pathway, as found in many other studies [38,39]. In addition, a WRKY TF gene (LITCHI018382) was detected as a highly correlated gene to the target genes, and the presence of the W-box element in the promoters of the target genes indicated that the WRKY TF may be a potential regulator gene. Interestingly, the two *MYB* genes identified (named as LcMYBPA1/2) had a greater similarity to the various PA biosynthesis pathway-specific MYBs (e.g., VvMYBPA1/2, DkMYB2/4, AtTT2) than to the anthocyanin biosynthesis-specific MYBs (e.g., MdMYB1/10, AtPAP1, SlANT1, Sl AN2) (Figure 7C), suggesting that the two MYB genes may be key regulators for the PA biosynthesis pathway in litchi.

## 3. Discussion

Litchi pericarp browning has been a key topic in the study of the postharvest characteristics of litchi fruit. Our recent studies showed that laccases played an important role in the rapid browning of litchi pericarp after harvest [4] because of their capacity to catalyze the oxidative polymerization of epicatechin (EC) and various procyanidins (PCs) [5] and that these browning-related proanthocyanidin (PA) substances were abundant in litchi pericarp [12]. In this study, we integrated PA metabolite profiling and transcriptome analysis of litchi pericarp with the stages of fruit development. Potential key structure genes and regulatory genes functioning in the PA biosynthesis pathway were identified.

### 3.1. PAs Highly Accumulate in the Young Fruit Stage and Decline with Fruit Development

Numerical studies and our previous study show that the litchi pericarp contain high levels of PAs [5,12,14,40]. In this study, the PA content of litchi pericarp was analyzed at four development stages, from young to fully mature fruit. We found that litchi pericarp mainly contained EC and PC A2/B1/B2, and that EC was the most abundant phenolic component in the pericarp. EC and PCs belong to PAs, also called condensed tannins, which are phenolic secondary metabolites that contribute to the protection of plant tissues. In leaves, PAs are mainly accumulated in the epidermal layers to provide protection from UV/strong light [41,42]. They are also considered to be defensive chemicals against various herbivores and fungal pathogens [23,42,43].

Our data showed that EC/PCs were synthesized in large quantities at the young fruit stage (30 DAF) and that the content decreased with fruit maturation. In grapes (*Vitis vinfera*), intensive PA accumulation in the berry skin was similarly found to occur in the stages immediately after fruit-set and then decrease during ripening [44]. In bilberry (*Vaccinium myrtillus* L.), PAs are abundant in the fruit and the amount of procyanidin subunits decrease during fruit ripening [45]. These findings indicate that high levels of these PA substances may serve as protectants during young fruit development. As the fruits develop, other protective mechanisms also participate, and the dependence on PAs as protectants may decline. For example, the cuticular waxes of litchi increase during the young stage and peak at 60 DAF, which may provide light protection or serve as a barrier against attack by insects or microorganisms [46,47]. In addition, anthocyanins accumulate in litchi pericarp after 75 DAF, and these also fulfil a similar protection role as the PAs [48]. Therefore, as litchi fruits mature, the dependence on PAs for protection may decline. However, it is worth noting that, although the EC/PC content decreases with fruit maturation, litchi pericarp still contains relatively high levels of these substances at the full maturation stage compared to the pericarp tissues of other fruits, such as apple [15,49] and grape [50]. These EC/PC substances in litchi pericarp could serve as substrates in LAC-mediated pericarp browning after fruit harvest [5].

### 3.2. Identification of Key Genes for PA Biosynthesis in Litchi Pericarp

Anthocyanin biosynthesis in litchi fruit has been studied extensively, but little is known about the synthesis of catechins [51,52]. In this study, transcriptome data analysis showed that flavonoid/PA biosynthesis pathways were active during litchi fruit development. Genes participating at each step of the flavonoid/catechin biosynthesis pathway were found in the RNA-Seq dataset, including 43 early or late structure genes (*PAL*, *4CL*, *CHS*, *CHI*, *F3H*, *DFR*, *ANS*, *LAR*, *ANR*). Half of these genes showed a significantly positive correlation with PA accumulation in the pericarp (Figure 4), indicating that multiple structure genes in the flavonoid/PA pathway are synergistically upregulated for the intensive accumulation of PAs. *ANR*, *LAR,* and *ANS* are the late structure genes functioning in the PA biosynthesis that branches from the flavonoid pathway [53]. LAR and ANR activities have been observed in many plants, and their activities are usually positively correlated with the accumulation of catechins [23,54]. Studies showed that the accumulation of PAs is highly correlated with the expression levels of *ANR*, *LAR,* and *ANS* genes in apple [55], tea tree [56], and grape [53,57]. In this study, we identified the transcripts of five *ANR*, two *LAR,* and two *ANS* genes in litchi pericarp during fruit development. The expression patterns of four genes (*ANR1a*, *LAR1*/*2,* and *ANS1*) were highly consistent with the catechin content trend, indicating that these four genes may play an important role in the synthesis of catechins in litchi pericarp.

In plants, the anthocyanin and PA biosynthesis pathways are both derived from the flavonoid biosynthetic pathway and share the chalcone–flavonol pathway, with common precursors and enzymes. Potential competition may exist between the biosynthesis of anthocyanins and PAs [6]. *ANS* and UDP-flavonoid glucosyltransferase (*UFGT*) are key anthocyanin biosynthetic genes in lichi fruit [53,54]. In this study, one *UFGT* gene (LITCHI002457) was upregulated after 75 DAF, which was inconsistent with the accumulation of anthocyanins. However, we found that *ANS1* was highly expressed at the young fruit stage when no anthocyanin was synthesized (Figure 1 and Figure 4). It is known that the function of *ANS* genes is required for both PA and anthocyanin biosynthesis [58] (Figure 4A). However, the accumulation of PAs or anthocyanins may be affected by the behavior of the genes of both pathways [18]. In this study, the high expression of *ANR1* in the young litchi fruit may lead to the dominant metabolite flow to the biosynthesis of PA, whereas, at the maturity stages, the upregulation of *UFGT* and downregulation of *ANR1a* may lead to a shift in metabolite flow from EC biosynthesis to anthocyanins (Figure 4). Similar results were found in grapes, with genes encoding enzymes required for both anthocyanin and PA biosynthesis being expressed prior to veraison when anthocyanins are not synthesized [44]. These results indicate that a shift from PAs to anthocyanins during litchi fruit development can be accomplished by a fine temporal regulation of individual pathway key genes.

### 3.3. Candidate Transcription Factors Involved in the Regulation of Catechin Biosynthesis in Litchi Pericarp

Many TFs are known to be involved in the regulation of the PA biosynthesis pathway in plants, including members of the MYB, bHLH, WRKY, and WD40 families [38,39]. Of these, the MYB family transcription factors are emerging as key players in the regulation of the anthocyanin and PA branches of the flavonoid pathway, via the formation of a functional ternary complex with bHLH and WD40, named MYB-bHLH-WD40 (MBW) [43,59,60]. A subset of MYB TFs have been identified as specifically regulating the PA pathway, including Arabidopsis MYB transcription factor TRANSPARENT TESTA2 (TT2) [30], grape VvMYBPA1 [32] and VvMYBPA2 [31], persimmon DkMYB2 and DkMYB4 [33,34], poplar (PtMYB134) [35], and *Medicago truncatula* (MtPAR) [36]. Some MYBs even promote the PA biosynthesis branch while repressing anthocyanin biosynthesis. For example, in grape berries, VviMYB86 upregulated two LAR genes, but downregulated the transcript levels of VviUFGT [61]. In this study, in litchi pericarp, two *MYB* genes (*MYBPA1*/*2*, Figure 7A,C) were highly correlated with *ANR1a*/*LAR1*/*ANS1* genes, and the existence of MYB binding elements in the promoter regions of these structure genes strongly indicates that the two MYBPAs may be key regulators of PA biosynthesis. Interestingly, in the phylogenetic analysis of plant MYBs, the two litchi MYBPAs were grouped in a subset of MYBs specific to the PA pathway, as described above, rather than in a set of MYBs for anthocyanin pathway regulation, such as MdMYB10 [62] and AtPAP1 [63] (Figure 7). Combined with the findings of previous studies, we suggest that the PA and anthocyanin pathways in plants may be individually regulated by specific subsets of MYBs. The individual regulation mechanism is plausible for the temporally different expression patterns of PA biosynthesis genes (*ANR* and *LAR*) and the key anthocyanin pathway gene (*UFGT*) in litchi pericarp (Figure 4), which may consequently lead to the temporally different accumulation patterns of PAs and anthocyanins (Figure 1).

In addition, WRKY TF was also found to interact with MYBs and function in the regulation of the PA pathway [64]. Here, we also found a WRKY gene that was highly correlated to *ANR1a*/*LAR1*/*ANS1* genes, and that the promoters of the target genes contained a WRKY binding element, W-box. In addition to MYBs, bHLHs, and WRKYs, other TF genes, such as NAC, C2H2, and AP2, whose functions in PA pathway regulation were barely understood, were also identified by the screening of the RNA-seq data. Therefore, it will be important to investigate whether and how the TFs identified in this study regulate the PA biosynthesis pathway in litchi fruit, in which such high levels of PAs accumulate.

## 4. Materials and Methods

### 4.1. Plant Materials

Litchi fruit (*Litchi chinensis* cv. “Huaizhi”) at four fruit development stages (30, 60, 75 and 85 days after female flower anthesis (DAF)) were obtained from an orchard on the campus of the South Agricultural University, Guangzhou city, Guangdong Province, China (23°7′ N, 113° E), and transported to the laboratory within 0.5 h of harvesting. The fruits were selected for uniformity of shape, size, and color. Fruit pericarp samples were collected and frozen in liquid nitrogen and then stored at −80 °C until further analysis.

### 4.2. Extraction and Analysis of PAs and Anthocyanins in the Pericarp

The extraction and HPLC analysis of soluble PAs and anthocyanins in each litchi pericarp sample were conducted according to methods described previously by [17,65], with some modifications. A quantity of 0.1 g litchi pericarp lyophilized powder was first extracted with 1.0 mL *n*-hexane solution containing 3% (*v*/*v*) formic acid and 1% (*w*/*v*) butylated hydroxytoluene (BHT) to remove fat and chlorophyll by vortexing for 10 min. After the centrifugation of 10,000 g for 10 min at 4 °C, the *n*-hexane was decanted and removed under reduced pressure for 30 min. After the comparison of the yields between the extraction achieved by 75% (*v*/*v*) acetone and methanol, we found that the yields achieved by the two extraction methods were similar (Appendix A). Accordingly, the substances in the residue were extracted with 1.5 mL methanol with 3% (*v*/*v*) formic acid and 1% (*w*/*v*) BHT. After vortexing for 2 min three times and sonication at 0 °C for 30 min, the supernatant of the extraction mixture was obtained after the centrifugation of 10,000 *g* for 10 min at 4 °C and subsequently filtered using a membrane (0.22 μm polyvinylidenedifluoride membrane, ANPEL Scientific Instruments, Shanghai, China). The residue was re-extracted using the abovementioned procedure, and the supernatant obtained was combined with the first one. The pericarp extract was subjected to HPLC analysis using a detector of Agilent 1260 InfinityⅡ detector (HPLC, Agilent Tech., Santa Clara, CA, USA). Then, 20 μL of the supernatant was injected into a ZORBAX Eclipse XDB C18 column (5 μm, 150 × 4.6 mm, Agilent, Santa Clara, CA, USA) and separated with a linear gradient of 0.2% (*v*/*v*) formic acid aqueous solution and methanol for 70 min at a flow rate of 0.5 mL min^−1^. The elution program was as follows: solvent A (methanol) and solvent B [0.2% (*v*/*v*) formic acid in water]; gradient, 0–15 min, 17–21% A; 15–28 min, 21% A; 28–32 min, 21–26% A; 32–45 min, 26–27% A; 45–60 min, 27–31% A; 60–70 min, 31–53% A; 70–80 min 53–90% A; 80–85 min 90–17% A; 85–90 min 17% A. The temperature of the column was 25 °C and the monitoring wavelengths were 280 nm (PAs) and 510 nm (anthocyanins). The PAs and anthocyanins were identified by a comparison with the standard retention time and absorption spectra including EC, PC A2/B1/B2, and cyanindin-3-rutinside (Cy3R) (all from Sigma-Aldrich, Saint Louis, MO, USA),, and confirmed using an internal standard method and spectrum comparison [17] (Appendix A). The standard HPLC chromatograms in Figure 1 were achieved with mixed standards with 0.3 mg/mL of EC and PC A2/ B2 and 0.75 mg/mL of Cy3R (Figure 1B). In addition, the contents of individual PAs and Cy3R were calculated according to the standard curves built using the standard concentration range (0. 5–2.5 mg/mL for EC; 0.25–1.25 mg/mL for PC A2; 0.2–1.5 mg/mL for PC B2 and 0.2–1.2 mg/mL for Cy3R), which showed a good linear relationship (R^2^ > 0.99) between the peak areas and the contents of the standards (Appendix A). Three biological replications were taken for each time point.

### 4.3. RNA Preparation and Transcriptome Analysis

Total RNA was extracted from the litchi pericarp using a Trizol reagent kit (Invitrogen, Carlsbad, CA, USA). The enriched mRNA was fragmented into short fragments using a fragmentation buffer and reversely transcribed into cDNA using the NEBNext Ultra RNA Library Prep Kit (NEB #7530, New England Biolabs, Ipswich, MA, USA). The cDNA libraries were constructed and sequenced using an Illumina Hiseq2500 (Gene Denovo Biotechnology Co., Ltd. Guangzhou, China). To obtain high quality clean reads, the reads were further filtered using Fastp (version 0.18.0) to remove “dirty” data, including low-quality reads, adaptor sequences, and rRNA reads. The clean reads were mapped to the litchi reference genome (http://121.37.229.61:82), and a novel genes prediction was performed using HISAT2. 2.4 [66]. The transcriptome analysis was performed using three biological replicates.

The total number of reads per kilobase per million mapped reads (RPKM) of each gene was calculated based on the length of the gene and the counts of the reads mapped to this gene. The differentially expressed genes (DEGs) were identified using the following parameters: a corrected *p*-value of 0.05 and an absolute fold change ≥ 2. Gene ontology (GO) annotation was implemented using Blast2GO software, and KOBAS (2.0) software was used for a Kyoto Encyclopedia of Genes and Genomes (KEGG) enrichment analysis of the DEGs. A principal component analysis (PCA) was performed using R package models (http://www.r-project.org/, accessed on 25 July 2022) to reveal the relationship of the samples. The sequence data in this study have been deposited in the NCBI SRA under accession number PRJNA883943.

A heatmap of PA biosynthesis-related genes was produced using the “pheatmap” package in R version 4.1.0. A correlation analysis of the expression of PA-related genes and EC/PC content was performed using Pearson correlations at a significance level of *p* < 0.05 (two-tailed). The expression patterns of the genes were hierarchically clustered using CLuster in Tbtools. The correlation between PA biosynthesis target genes and transcription factor (TF) genes was acquired using the R package of the weighted correlation network analysis (WGCNA) [67].

### 4.4. Bioinformatic and Phylogenetic Analyses of ANR, LAR and ANS Genes

The SignalP 4.1 server was used to predict the presence and location of signal peptide cleavage sites in amino acid sequences (http://www.cbs.dtu.dk/services/SignalP-4.1/, accessed on 25 June 2022). The NetNglyc server was applied for the N-Glycosylation site prediction (https://services.healthtech.dtu.dk/service.php?NetNGlyc-1.0, accessed on 25 June 2022). A phylogenetic analysis based on amino acid sequences was performed using MEGA (version 5.02) and the neighbor-joining method with 1000 bootstrap replicates. The ANR, LAR, and ANS sequences from various plant species used in the phylogenetic tree analysis are listed in Appendix A.

### 4.5. Quantitative Real-Time PCR (qRT-PCR) Analysis

To verify the RNA-Seq results, five *ANR*, two *ANS,* and two *ANR* genes potentially involved in the biosynthesis of catechins were selected for qRT-PCR analysis. The qRT-PCR analysis was performed according to the method developed by [5]. The qRT-PCR was conducted using a Bio-Rad CFX96 system (Bio-Rad, Hercules, CA, USA) in combination with the Bio-Rad iTaq Universal SYBR Green Supermix (Bio-Rad, USA). Cycling conditions included an initial denaturation at 95 °C for 1 min, followed by 40 cycles at 95 °C for 15 s, 57 °C for 30 s, and 72 °C for 35 s. Gene-specific primers for qRT-PCR analysis are shown in Appendix A. *LcActin* (LITCHI007623) was used as an internal reference to normalize the expression levels of the target genes in different samples. The expression levels of the selected genes were calculated using the 2^–ΔΔCt^ method [68]. Three technical replicates and three biological replicates were analyzed for each mRNA sample, and each time point included three biological replicate samples.

### 4.6. Analysis of Cis-Acting Regulatory Elements in the Promoter

The 2 kb upstream sequences of the *ANR*, *LAR,* and *ANS* gene translation initiation codon were downloaded from the litchi reference genome database. Using the PlantCare database (http://bioinforma-tics.psb.ugent.be/webtools/plantcare/html/, accessed on 5 August 2022), cis-regulatory elements in the 2 kb upstream regions were predicted.

### 4.7. Statistical Analysis

Statistical analyses of the data were performed using SPSS 19.0 software (International Business Machines Corporation, Chicago, IL, USA). One-way analysis of variance (ANOVA) was used for significance analysis. Data are presented as the mean ± standard error (SE). There were three replications for the analysis of each parameter.

## 5. Conclusions

Nine PA biosynthesis pathway-related genes (five *ANR*, two *LAR,* and two *ANS* genes) were screened in the transcriptome data of litchi pericarp during fruit development. Of these, *ANR1a, LAR1,* and *ANS1* showed highly positive correlations with changes in the content of the major PA substances, including EC and several PCs, suggesting that they are potentially the key PA biosynthesis genes in litchi pericarp. Multiple TF genes, including MYB, bHLH, WRKY, and AP2 family members, which were highly correlated with *ANR1a, LAR1,* and *ANS1* genes, are likely to be regulatory TF genes in the PA biosynthesis of litchi pericarp. The identification of the key PA biosynthesis structure and their regulatory genes sheds light on the molecular mechanism relating to the accumulation of high levels of PA substances in litchi pericarp.

## Figures and Tables

**Figure 1 ijms-24-00532-f001:**
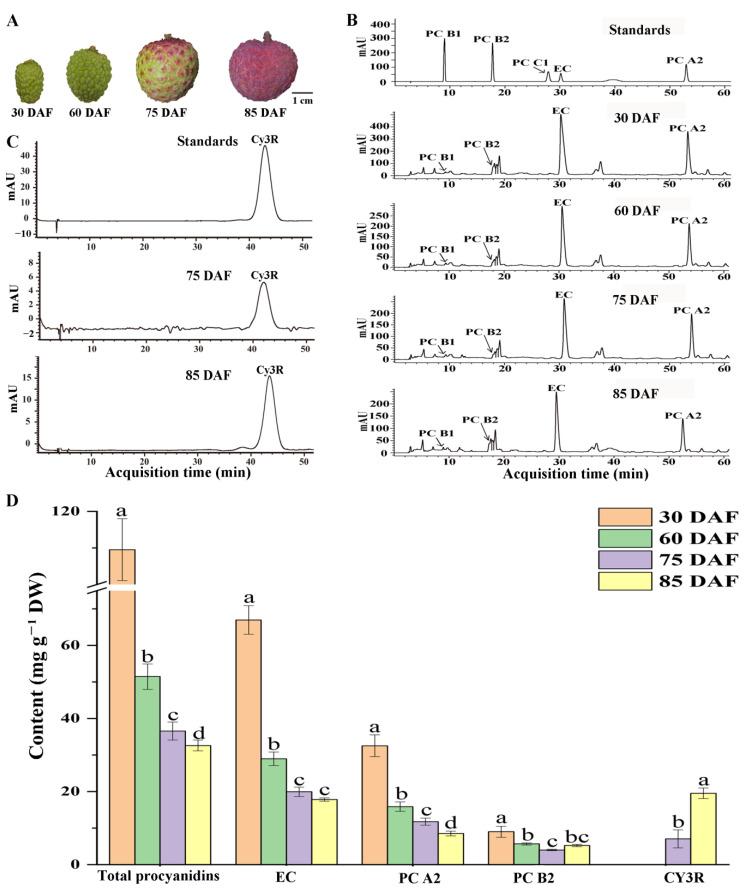
Proanthocyanidin and anthocyanin content of litchi pericarp. (**A**) Images of litchi fruit during development at 30, 60, 75, and 85 DAF, representing the young, growing, color-break, and mature stages, respectively. (**B**) Mixed standard HPLC chromatograms of a variety of (epi)catechin/PCs and the soluble PA extract of litchi pericarp samples at the four stages as described in (**A**) with the absorbance at 280 nm. (**C**) Standard HPLC chromatograms of Cy3R and the pericarp extracts (75 and 85 DAF) with the absorbance at 510 nm. (**D**) Epicatechin/PC and anthocyanin content of litchi pericarp at the four developmental stages. Values are the means of three biological repeats. Error bars indicate ± SE. Statistical significance was determined using *t-tests*, and significant differences (*p* < 0.05) among the samples at different developmental stages were classified with the letters a, b, c, or d. EC: (−)-epicatechin; CT: (+)-catechin; PC: procyanidin; and Cy3R: cyanidin-3-O-rutinoside.

**Figure 2 ijms-24-00532-f002:**
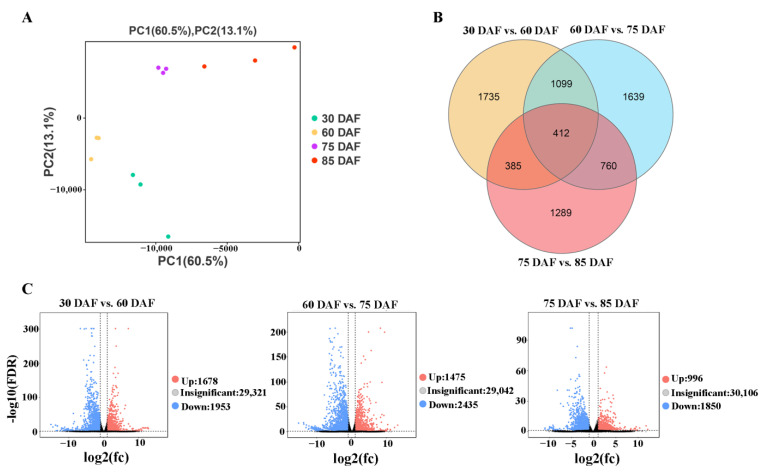
Profiling the DEGs in the pericarp during litchi fruit development. (**A**) 2D PCA score plot of the samples at the four development stages of 30, 60, 75, and 85 DAF, as indicated in Figure 1A. (**B**) Venn diagram of the DEGs of the 30 DAF vs. 60 DAF, 60 DAF vs. 75 DAF, and 75 DAF vs. 85 DAF comparison pairs. (**C**) Volcano plots of the DEGs of 30 DAF vs. 60 DAF, 60 DAF vs. 75 DAF, and 75 DAF vs. 85 DAF comparison pairs. Black dots represent nonDEGs, red dots represent upregulated DEGs, and green dots represent downregulated DEGs.

**Figure 3 ijms-24-00532-f003:**
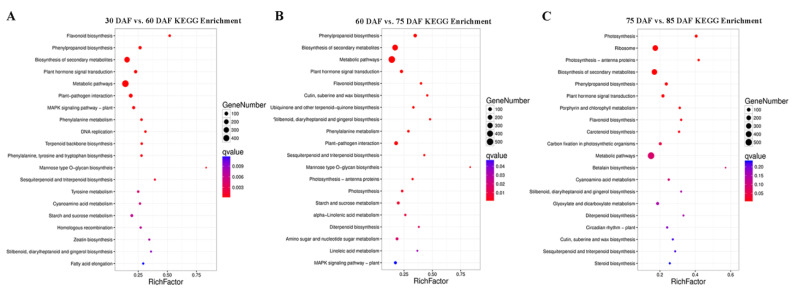
KEGG pathway enrichment analysis of DEGs during litchi fruit development. The 20 most significant catalogues with the lowest corrected q-values based on the DEGs by (**A**) 30 DAF vs. 60 DAF, (**B**) 60 DAF vs. 75 DAF, and (**C**) 75 DAF vs. 90 DAF comparison. The number of DEGs is indicated by the size of the circle, and the circle from red to blue represents the q-values.

**Figure 4 ijms-24-00532-f004:**
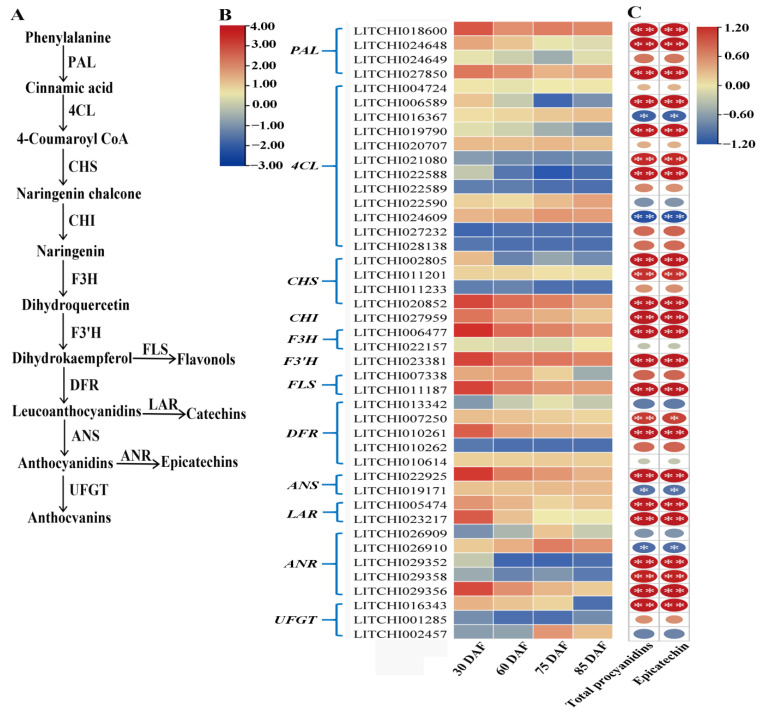
Scheme of the proanthocyanidin biosynthesis pathway and correlation of flavonoid/proanthocyanidin biosynthesis-related gene expression and EC/PC accumulation during litchi fruit development. (**A**) Enzyme names are abbreviated as follows; PAL, phenylalanine ammonia lyase; 4CL, 4 coumarate CoA ligase; CHS, chalcone synthase; CHI, chalcone isomerase; F3H, flavanone 3-hydroxylase; F3′H, flavanone 3′-hydroxylase; DFR, dihydroflavonol reductase; FLS, flavonol synthase; ANS, anthocyanidin synthase; UFGT, UDP-flavonoid glucosyltransferase; ANR, anthocyanidin reductase; and LAR, leucoanthocyanidin reductase. (**B**) Heatmap of the genes involved in flavonoid/proanthocyanidin biosynthesis in the fruit development stages, as described in Figure 4A. The color from blue to red represents low to high gene expression levels, based on the RNA-seq analysis. Colors indicate relative abundance, ranging from blue (low) to red (high). (**C**) Pearson correlation of the gene expression levels in (**B**) and EC and total PC content. The size and color of the circles indicate the correlation coefficient, ranging from small (low) to large (high) and red (positive correlation) to blue (negative correlation), respectively. “*” and “**” in ellipses represent significant correlation (*p* < 0.05) and extremely significant difference (*p* < 0.01), respectively.

**Figure 5 ijms-24-00532-f005:**
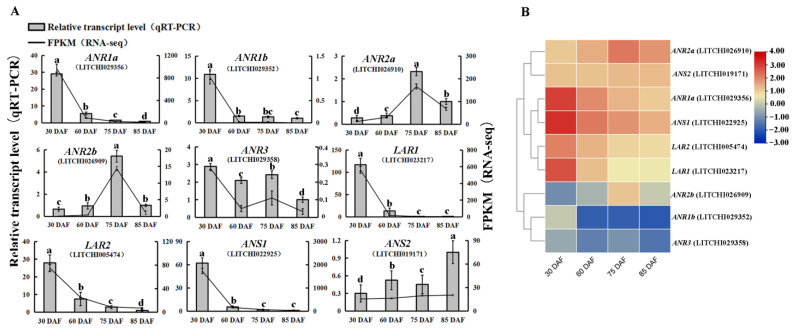
Verification and hierarchical clustering of the expression of PA biosynthesis-related genes. (**A**) qRT-PCR analysis of nine PA biosynthesis-related genes in litchi pericarp during four development stages, including *ANR1a* (LITCHI029356), *ANR1b* (LITCHI029352), *ANR2a* (LITCHI026910), *ANR2b* (LITCHI026909), *ANR5* (LITCHI029358), *LAR1* (LITCHI023217), *LAR2* (LITCHI005474), *ANS1* (LITCHI022925), and *ANS2* (LITCHI019171). The gene expression was determined by fold change relative to the reference gene (*Actin*, LITCHI007623). Values are means of three biological repeats. Error bars indicate ± SE. Statistical significance was determined using *t-tests*, significant difference (*p* < 0.05) among the samples at different developmental stages were classified with the letter a, b, c, or d. (**B**) Heatmap of the expression of catechin biosynthesis-related genes (*ANR*, *LAR*, *ANS*). The expression patterns of the genes were hierarchically clustered using TBtools CLuster. The heatmap was generated as described in Figure 4B.

**Figure 6 ijms-24-00532-f006:**
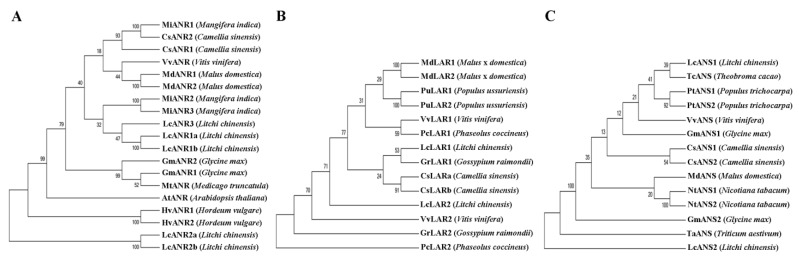
Rooted neighbor-joining phylogenetic trees of the full-length amino acid sequences of ANR (**A**), LAR (**B**), and ANS (**C**) proteins. Five ANRs, two LARs and two ANSs of litchi, as indicated in Figure 4B, were subjected to phylogenetic analysis with the characterized relevant proteins from other plant species based on published data. The accession numbers of these proteins are listed in Appendix A. The phylogenetic trees were constructed using the maximum likelihood method with 1000 bootstrap replicates using MEGA 5.02.

**Figure 7 ijms-24-00532-f007:**
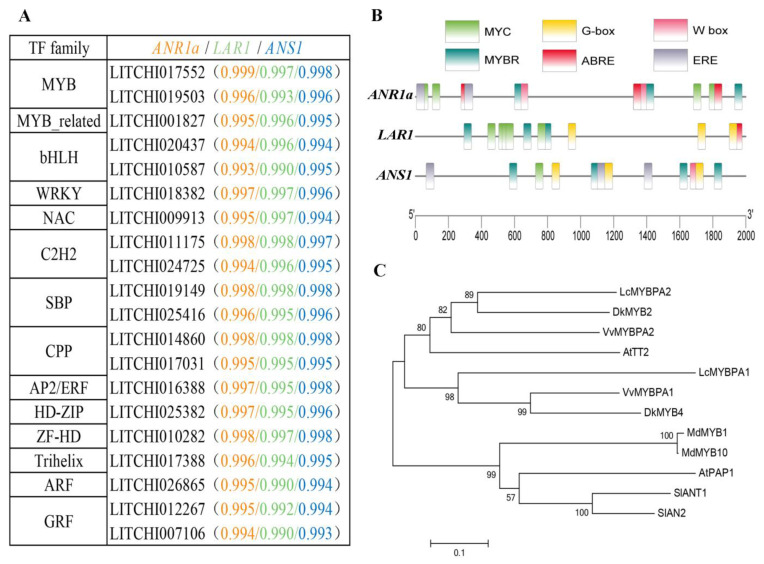
(**A**) The top 20 transcription factors (TFs) showing high correlation with PA biosynthesis candidate genes (*ANR1a*, *LAR1,* and *ANS1*). (**B**) Cis-regulatory elements detected in 2000 bp upstream of *ANR1a* /*LAR1*/*ANS1* genes based on PlantCARE analysis. (**C**) Rooted neighbor-joining phylogenetic trees of the full-length amino acid sequences of MYB proteins. The accession numbers of AtTT2 and AtPAP1 from *Arabidopsis thaliana*, VvMYBPA1/2 (*Vitis vinifera*), DkMYB2/4 (*Diospyros kaki*), LcMYBPA1/2 (*Litchi chinensis*), MdMYB1/10 (*Malus domestica*), and SlANT1 and SlAN2 (*Solanum lycopersicum*) are listed in Appendix A.

**Table 1 ijms-24-00532-t001:** Prediction of the biochemical characteristics of ANR, LAR, and ANS proteins.

Gene ID	Gene Name	Number of Amino Acids	Molecular Weight (kD)	Isoelectric Point	Signal Peptide	N-Glycosyl Sites
LITCHI029356	*LcANR1a*	336	36.45	6.55	No	2
LITCHI029352	*LcANR1b*	334	36.3	6.55	No	2
LITCHI026910	*LcANR2a*	349	38.89	5.72	No	2
LITCHI026909	*LcANR2b*	343	38.01	5.35	No	3
LITCHI029358	*LcANR3*	281	30.18	8.87	No	2
LITCHI023217	*LcLAR1*	357	39.54	5.77	No	1
LITCHI005474	*LcLAR2*	343	38.05	6.13	No	3
LITCHI022925	*LcANS1*	357	40.56	5.59	No	1
LITCHI019171	*LcANS2*	345	38.29	6.21	No	2

## Data Availability

NCBI BioProject online repository (accession number: https://www.ncbi.nlm.nih.gov/bioproject/PRJNA883943/, accessed on 25 September 2022).

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
