# Peer review of "Metabolite and Transcriptome Profiles of Proanthocyanidin Biosynthesis in the Development of Litchi Fruit"

_ijms, 2022, doi:10.3390/ijms24010532_

Round 1
Reviewer 1 Report
According to the authors, the main objective of the manuscript entitled “Integrative Analysis of Metabolite Profiles and Transcriptomes Reveals Key Proanthocyanidin Biosynthesis Structure and Regulatory Genes in the Development of Litchi Fruit” is to correlate PA metabolite profiles with transcriptome changes in the pericarp during litchi fruit development.
The topic is relevant and within the scope of International Journal of Molecular Sciences journal. Both the approach and technology are sound. Appropriate repetitions were conducted at each step of the procedures to satisfy the statistical analysis. In addition, the manuscript is clearly written and easy to follow.
There are only a few points that could be considered.
-Some abbreviation were mentioned in text without clarifying their full meaning like bHLH, WRKY, AP2, NAC, C2H2, SBP, MYR, MYC, ADE/LAC…
-Line 448, correct the writing of (Figureure 5).
-Line 469 kindly mention the duration of extraction of litchi pericarp using methanol.
-Line 475 what was the ratio between aqueous solution and methanol used for linear gradient in HPLC analysis?
-Line 480, please state the specific range of concentration for each standard used in HPLC analysis.
-The conclusion part does not contain any future aspects.
Author Response
Response to Reviewer 1 Comments
Dear reviewer,
Thank you very much for reviewing our manuscript ijms-2067386. Your professional comments and suggestions made great help to improve the quality of this manuscript. According to your comments, we have done some revisions.
Reviewer 1:
According to the authors, the main objective of the manuscript entitled “Integrative Analysis of Metabolite Profiles and Transcriptomes Reveals Key Proanthocyanidin Biosynthesis Structure and Regulatory Genes in the Development of Litchi Fruit” is to correlate PA metabolite profiles with transcriptome changes in the pericarp during litchi fruit development.
The topic is relevant and within the scope of International Journal of Molecular Sciences journal. Both the approach and technology are sound. Appropriate repetitions were conducted at each step of the procedures to satisfy the statistical analysis. In addition, the manuscript is clearly written and easy to follow.
There are only a few points that could be considered.
Point 1: Some abbreviation were mentioned in text without clarifying their full meaning like bHLH, WRKY, AP2, NAC, C2H2, SBP, MYR, MYC, ADE/LAC…
Response 1: According to the suggestion, all abbreviations have been checked to be with their full meaning when they appear in the manuscript for the first time.
Point 2: Line 448, correct the writing of (Figureure 5).
Response 2: Thanks for the comment. It has been corrected in the manuscript (line 542).
Point 3: Line 469 kindly mention the duration of extraction of litchi pericarp using methanol.
Response 3: Thanks for the comment. The substances in the pericarp powder were extracted with 1.5 mL methanol with 3% (v/v) formic acid and 1% (w/v) BHT. We extracted the substances by vortex for 2 min for three times and sonication at 0 ℃ for 30 min. The above extraction procedure was repeated twice, and the supernatant was combined. The details of PA extraction of litchi pericarp using methanol have been added in the manuscript (line 572–578).
Point 4: Line 475 what was the ratio between aqueous solution and methanol used for linear gradient in HPLC analysis?
Response 4: Thanks for the question. The elution program was as follows: solvent A (methanol) and solvent B [0.2% (v/v) formic acid in water]; gradient, 0−15 min, 17−21% A; 15−28 min, 21% A; 28−32 min, 21−26% A; 32−45 min, 26−27% A; 45−60 min, 27−31% A; 60−70 min, 31−53% A; 70−80 min 53−90% A; 80−85 min 90−17% A; 85−90 min 17% A. The details have been added in the manuscript (line 586–590).
Point 5: Line 480, please state the specific range of concentration for each standard used in HPLC analysis.
Response 5: Thanks for the comment. In our revised manuscript, we indicate the specific range of concentration for each standard in the HPLC method description. We also show the standard curves in added supplementary Figure 5S (line 599–602).
Point 6: The conclusion part does not contain any future aspects.
Response 6: Thanks for the comment. We have added future aspects in the conclusion part of revised manuscript (line 683–686).

Reviewer 2 Report
The manuscript is well-written and highlights the work done by the authors. However, the figures' quality should be improved and I would recommend maintaining the standard order of the manuscript structure (introduction; material and methods; results; discussion and conclusion) aiming to facilitate reading.
Author Response
Response to Reviewer 2 Comments
Dear reviewer,
Thank you very much for reviewing our manuscript ijms-2067386. Your professional comments and suggestions made great help to improve the quality of this manuscript. According to your comments, we have done some revisions.
Reviewer 2:
Point 1: The manuscript is well-written and highlights the work done by the authors. However, the figures' quality should be improved and I would recommend maintaining the standard order of the manuscript structure (introduction; material and methods; results; discussion and conclusion) aiming to facilitate reading.
Response 1: We appreciate for your comments and suggestions. We used the Microsoft Word template of instructions for authors to prepare the manuscript, in the order of the text (1. Introduction; 2. Results; 3. Discussion; 4. Materials and Methods; 5. Conclusions). We also tried our best to improve the figures' quality. For example, we improved the HPLC chromatography profiles in the revised Figure 1. The pathway in the former Figure 1 was integrated into the revised Figure 4, etc.

Reviewer 3 Report
The manuscript entitled “Integrative Analysis of Metabolite Profiles and Transcriptomes Reveals Key Proanthocyanidin Biosynthesis Structure and Regulatory Genes in the Development of Litchi Fruit” and authored by Ruihao Zhong, Junbin Wei, Bin Liu, Honghui Luo, Zhaoqi Zhang, Xuequn Pang, and Fang Fang, deals with the study of the proanthocyanidin profiles and transcriptome changes in the pericarp during litchi fruit development, with the aim to investigate the molecular mechanism of their accumulation and regulation.
The manuscript contains information that can seriously contribute to knowledge in this field. It appears well written and structured, although several typos are present in the main text. However, I do not feel that this would be a problem that would compromise its publication in IJMS.
However, some revisions need to be made before I can consider this manuscript suitable for publication in the journal. Below is a series of comments, listed point by point:
AFFILIATION INFORMATION: In the affiliations section, the email should be listed for each author along with the acronym in brackets. The acronym assigned in this section, should be the same one used later for the contributions section.
ABSTRACT: The abstract is too long. It should be composed of a single section consisting of no more than 200 words. This Reviewer understands the difficulty in writing such a short abstract, especially in a scientific article where several data are presented. However, I strongly advise authors to follow the journal rules.
KEYWORDS: The keywords should be completely changed. The utility of these terms is to facilitate the search of the article using common scientific search engines (PubMed, GoogleScholar, Scopus, etc.), which rely on the terms contained in title, abstract, and keywords. Consequently, using terms that are already in these sections as keywords is inappropriate. I strongly suggest that the keywords be changed before re-submission and add new ones (max 10).
INTRODUCTION: The introduction is well organized and structured. It begins with a brief description of the fruit that is the subject of the study and then expands on PACs. However, little is described about this molecules. In particular, the authors should point out the fact that (i) PACs are plant secondary metabolites that are limitedly distributed in the plant kingdom, and only some plants have the ability to produce them; (ii) they have important functions in the plant, outside of ripening; and (iii) they find strong interest in the scientific field, because it has been proven that consumption of fruits rich in PACs has positive effects on human health. The authors can find much of this information in this recent review: 10.3390/antiox10081229.
I don't think Figure 1 is essential in the introduction. The authors should consider removing it or moving it as a supplementary file.
MATERIALS AND METHODS: This section is fairly well written, although some missing features are present.
LINE 461: Please provide the geographic coordinates of the orchard;
SECTION 4.2: The best extraction method for PACs is to use 75% acetone acidified to 0.5% with Acetic Acid (10.3390/antiox10081229; 10.1021/jf9015398). I do not think it is a problem to use alternative methods for PACs extraction, my only doubt is related to the exhaustiveness of the reaction process. Have the authors evaluated the effectiveness of the extraction yield by, for example, repeating the extraction on the spent matrices? If yes, this detail should be included.
In addition, the part describing the HPLC part lacks important analytical information. For example, (i) what type of detector was coupled to the HPLC to detect PACs?; ii) Has the method used by the authors been previously validated? It is an unsuitable and uncommon method for the analysis of these metabolites, which are normally chromatographically separated by a gradient of water and acetonitrile; please, provide a reference if available (iii) the analysis of PACs by simple UV/Vis detection is not so simple. PACs of different degree of polymerization and with different linkage (A-type or B-type) are well known to co-elute into a single peak (10.3390/antiox10081229). Only methods using mass spectrometry are capable of detecting qualitative differences. The authors should specify in the main text that although they were detected through the use of standards, it is possible that other PACs other than those expected are present in those peaks (as among other things is evident from Figure 2, Panel B); (iv) Please provide LOD, LOQ, and ME of all using standards for quantitative analysis. This factor is very important, especially for compounds named by the authors PCB1, in order to prove that what the authors quantified is actually PACs and not background noise.
CONCLUSION: This section should be implemented. Unfortunately, unlike abstract section, it appears to be too short and lacking in information or achieved data. In addition, the authors should also include potential future perspectives.
Author Response
Response to Reviewer 3 Comments
Dear reviewer,
Thank you very much for reviewing our manuscript ijms-2067386. Your professional comments and suggestions made great help to improve the quality of this manuscript. According to your comments, we have done some revisions.
Reviewer 3
The manuscript entitled “Integrative Analysis of Metabolite Profiles and Transcriptomes Reveals Key Proanthocyanidin Biosynthesis Structure and Regulatory Genes in the Development of Litchi Fruit” and authored by Ruihao Zhong, Junbin Wei, Bin Liu, Honghui Luo, Zhaoqi Zhang, Xuequn Pang, and Fang Fang, deals with the study of the proanthocyanidin profiles and transcriptome changes in the pericarp during litchi fruit development, with the aim to investigate the molecular mechanism of their accumulation and regulation.
Point 1: The manuscript contains information that can seriously contribute to knowledge in this field. It appears well written and structured, although several typos are present in the main text. However, I do not feel that this would be a problem that would compromise its publication in IJMS.
Response 1: We sincerely appreciate the time and effort in providing constructive comments. For the problem of several typos, this manuscript has undergone English language editing by MDPI. The text has been checked for correct use of grammar and common technical terms, and edited to a level suitable for reporting research in a scholarly journal.
However, some revisions need to be made before I can consider this manuscript suitable for publication in the journal. Below is a series of comments, listed point by point:
Point 2: AFFILIATION INFORMATION: In the affiliations section, the email should be listed for each author along with the acronym in brackets. The acronym assigned in this section, should be the same one used later for the contributions section.
Response 2: Thanks for your suggestions. The information has been added in the revised manuscript (line 13–15).
Point 3: ABSTRACT: The abstract is too long. It should be composed of a single section consisting of no more than 200 words. This Reviewer understands the difficulty in writing such a short abstract, especially in a scientific article where several data are presented. However, I strongly advise authors to follow the journal rules.
Response 3: Thanks for your suggestions. We tried to shorten the abstract to 227 words.
Point 4: KEYWORDS: The keywords should be completely changed. The utility of these terms is to facilitate the search of the article using common scientific search engines (PubMed, GoogleScholar, Scopus, etc.), which rely on the terms contained in title, abstract, and keywords. Consequently, using terms that are already in these sections as keywords is inappropriate. I strongly suggest that the keywords be changed before re-submission and add new ones (max 10).
Response 4: Thanks for your suggestions. The keywords have been changed in the manuscript (litchi; proanthocyanidin biosynthesis; transcriptomic analysis; gene screening; gene expression regulation)
Point 5: INTRODUCTION: The introduction is well organized and structured. It begins with a brief description of the fruit that is the subject of the study and then expands on PACs. However, little is described about this molecules. In particular, the authors should point out the fact that (i) PACs are plant secondary metabolites that are limitedly distributed in the plant kingdom, and only some plants have the ability to produce them; (ii) they have important functions in the plant, outside of ripening; and (iii) they find strong interest in the scientific field, because it has been proven that consumption of fruits rich in PACs has positive effects on human health. The authors can find much of this information in this recent review: 10.3390/antiox10081229.
Response 5: Thanks for your suggestions. We have added relevant descriptions in the introduction (line78–82). PAs play multiple roles in plants, particularly have important functions in the protection from biotic and abiotic damage (Winkel-Shirley, 2001). Recently, the great values to human health of PAs, such as antioxidant, anticancer and cardiovascular disease alleviation, have attracted wide interest among researchers (Mannino et al., 2021).
Point 6: I don't think Figure 1 is essential in the introduction. The authors should consider removing it or moving it as a supplementary file.
Response 6: We completely accept your suggestion and combine Figure 1 and Figure 5 in the manuscript.
Point 7: MATERIALS AND METHODS: This section is fairly well written, although some missing features are present.
LINE 461: Please provide the geographic coordinates of the orchard;
Response 7: According to the suggestion, the information about the geographic coordinates of the orchard (23°7′ N, 113° E) have been added in the manuscript (line 560).
Point 8: SECTION 4.2: The best extraction method for PACs is to use 75% acetone acidified to 0.5% with Acetic Acid (10.3390/antiox10081229; 10.1021/jf9015398). I do not think it is a problem to use alternative methods for PACs extraction, my only doubt is related to the exhaustiveness of the reaction process. Have the authors evaluated the effectiveness of the extraction yield by, for example, repeating the extraction on the spent matrices? If yes, this detail should be included.
Response 8: We are sorry that we did not describe the extraction procedure clearly in our former manuscript. It is true that we used methanol [with 3% (v/v) formic acid and 1% (w/v) BHT] to extract PACs from litchi pericarp. Detailed extraction procedure, including the extraction times, had been added to the manuscript (line 570-581). You are right that acid 75% acetone is commonly applied for PA extraction from plant tissues. Beside 75% acetone extraction, methanol extraction was also applied for PA extraction from fruit tissue (Tsao et al., 2003) and from other plant tissues (Khallouki et al., 2007). We had compared the effectiveness of the extraction yield between the acetone and methanol extraction, and we found that the yield by the two methods were similar (Figure 4S). Since there is a big solvent (acetone) peak in the chromatography profile of the acetone-extract when using our HPLC separation condition (Figure 4S), we decide to use methanol extraction instead of acetone. We describe the yield comparison in the method part to indicate the reason to use methanol extraction, in the revised manuscript (line 571-573).
Point 9: In addition, the part describing the HPLC part lacks important analytical information. For example, (i) what type of detector was coupled to the HPLC to detect PACs?; ii) Has the method used by the authors been previously validated? It is an unsuitable and uncommon method for the analysis of these metabolites, which are normally chromatographically separated by a gradient of water and acetonitrile; please, provide a reference if available (iii) the analysis of PACs by simple UV/Vis detection is not so simple. PACs of different degree of polymerization and with different linkage (A-type or B-type) are well known to co-elute into a single peak (10.3390/antiox10081229). Only methods using mass spectrometry are capable of detecting qualitative differences. The authors should specify in the main text that although they were detected through the use of standards, it is possible that other PACs other than those expected are present in those peaks (as among other things is evident from Figure 2, Panel B); (iv) Please provide LOD, LOQ, and ME of all using standards for quantitative analysis. This factor is very important, especially for compounds named by the authors PCB1, in order to prove that what the authors quantified is actually PACs and not background noise.
Response 9: We appreciate for your comments and suggestions.
(i) The HPLC analysis used an Agilent 1260 Infinityâ…¡UV/Vis detector.
(ii) It is true that a gradient of water and acetonitrile is commonly used for HPLC separation of PACs. The HPLC separation by the gradient of water and methanol in the present study has been previously validated in our previous studies (Gong et al., 2018; Wei et al., 2021; Liu et al., 2022). As showed in Figure 1A, the mixed standards or the PA substances from litchi were well separated by the method. Methanol/water gradient was also used in the HPLC separation of PA substances extracted from other plant species (Khallouki et al., 2007; Szankowski et al., 2009).
(iii) Both UV/Vis and fluorimeter detector are used for PAC detection in HPLC systems. UV/Vis detector is a more common detector for HPLC and is applied by many studies of PAC analysis (Reichel et al., 2011; Li et al., 2012). We achieved excellent standard curves for EC, PC A2, PC B2 (R2>0.99) by using UV detection (Figure S5), indicating UV signal is well correlated to the amount of the substances. As you pointed out, in our previous studies, we also found that PACs of different degree of polymerization and with different linkage (A-type or B-type) co-eluted into a single peak (Gong et al., 2018; Wei et al., 2021). PC A2 is known to be abundant in litchi pericarp besides EC, and its peak of was verified with mass spectrometry by Reichel et al. (2011) and in our previous study (Gong et al., 2018). In the present study, the peaks of PC A2, PC B2 and EC were also verified by addition of standards in the extract (Figure 3S), and by the identity of the absorption spectra of the identified peaks to those of the respective standards (Figure 4S). Still, we agree with the reviewer that we should specify in the main text that “although they were detected through the use of standards, it is possible that other PACs other than those expected are present in those peaks (as among other things is evident from Figure 2, Panel B)”.
(iv) The standard curves were built with the standards at more or less the same range of concentration (0.25-2 mg/mL), even though the ranges were slightly different between the standards (Figure S5A). For EC, PC A2 and B2, the concentrations in the extract are within the range of the concentration for the standard curves (Figure S5B), and are supposed to be far above their LOD, LOQ values by our HPLC method. However, as you pointed out that some peaks might have backgrounds noise, e.g. PC B1. Since its concentration is quite low, and it is not a major PA for the total procyanidin contents. We decided to omit its content in the new figure 1A.
Point 10: CONCLUSION: This section should be implemented. Unfortunately, unlike abstract section, it appears to be too short and lacking in information or achieved data. In addition, the authors should also include potential future perspectives.
Response 10: Thanks for the comments. We have added more summary for the results and future potential future perspectives in the revised manuscript (line 683–686).
References:
- Winkel-Shirley, B. Flavonoid Biosynthesis. A colorful model for genetics, biochemistry, cell biology, and biotechnology. Plant Physiol. 2001, 126, 485-93, doi: 10.1104/pp.126.2.485.
- Mannino, G.; Chinigò, G.; Serio, G.; Genova, T.; Gentile, C.; Munaron, L.; Bertea, C.M. Proanthocyanidins and where to find them: a meta-analytic approach to investigate their chemistry, biosynthesis, distribution, and effect on human h Antioxidants. 2021, 10, 1229. doi: 10.3390/antiox10081229.
- Tsao, R.; Yang, R.; Young, J.C.; Zhu, H. Polyphenolic profiles in eight apple cultivars using high-performance liquid chromatography (HPLC). Agr. Food. Chem. 2003, 51, 6347–6353. doi: 10.1021/jf0346298.
- Khallouki, F.; Haubner, R.; Hull, W. E.; Erben, G.; Spiegelhalder, B.; Bartsch, H.; Owen, R. W. Isolation, purification and identification of ellagic acid derivatives, catechins, and procyanidins from the root bark of Anisophyllea dichostyla R. Br. FOOD CHEM TOXICOL. 2007, 45, 472-485. doi: 10.1016/j.fct.2006.09.011.
- Szankowski, I.; Flachowsky, H.; Li; Halbwirth, H.; Treutter, D.; Regos, I.; Hanke, MV.; Stich, K.; Fischer, TC. Shift in polyphenol profile and sublethal phenotype caused by silencing of anthocyanidin synthase in apple (Malus sp.). Planta. 2008, 229, 681-692. doi: 10.1007/s00425-008-0864-4.
- Reichel, M.; Carle, R.; Sruamsiri, P.; Neidhart, S. Changes in flavonoids and nonphenolic pigments during on-tree maturation and postharvest pericarp browning of litchi (Litchi chinensis Sonn.) as shown by HPLC-MSn. Agric. Food Chem. 2011, 59 , 3924-3939. doi: 10.1021/jf104432r.
- Li, W.; Liang, H.; Zhang, M.; Zhang, R.; Deng, Y.; Wei, Z.; Zhang, Y.; Tang, X. Phenolic profiles and antioxidant activity of litchi (Litchi Chinensis Sonn.) fruit pericarp from different commercially available cultivars. Molecules. 2012, 17, 14954-14967. doi: 10.3390/molecules171214954.
- Gong, Y.; Fang, F.; Zhang, X.; Liu, B.; Luo, H.; Li, Z.; Zhang, X.; Zhang, Z.; Pang, X. B type and complex A/B type epicatechin trimers isolated from litchi pericarp aqueous extract show high antioxidant and anticancer activity. J. Mol. Sci. 2018, 19, 301, doi:10.3390/ijms19010301.
- Wei, J.; Zhang, X.; Zhong, R.; Liu, B.; Zhang, X.; Fang, F.; Zhang, Z.; Pang, X. Laccase-mediated flavonoid polymerization leads to the pericarp browning of litchi fruit. J. Agr. Food Chem. 2021, 69, 15218-15230, doi:10.1021/acs.jafc.1c

Round 2
Reviewer 3 Report
The authors addressed all my concerns. The manuscript is now suitable as publication
Author Response
Response to Reviewer 3 Comments
Dear reviewer,
Thank you very much for reviewing our manuscript ijms-2067386. Your professional comments and suggestions made great help to improve the quality of this manuscript.